# INSTRUCTION-FINETUNED FOUNDATION MODELS FOR MULTIMODAL WEB NAVIGATION

**Hiroki Furuta**[1,2][*] **Ofir Nachum**[2] **Kuang-Huei Lee**[2] **Yutaka Matsuo**[1]
**Shixiang Shane Gu**[2,1] **Izzeddin Gur**[2]
[1]The University of Tokyo     [2]Google Research, Brain Team
`furuta@weblab.t.u-tokyo.ac.jp`

## ABSTRACT

We propose an instruction-aligned multimodal agent for *autonomous web navigation* – i.e., sequential decision making tasks employing a computer interface. Our approach is based on supervised finetuning of vision and language foundation models on a large corpus of web data consisting of webpage screenshots and HTML. Specifically, we use vision transformers on sequences of web page screenshots to extract patch-level image features. These features are concatenated with embedding of tokens in HTML documents. Using an instruction-finetuned large language model, we jointly encode both vision and HTML modalities and decode web actions such as *click* and *type*. We show that our method outperforms previous approaches by a significant margin, even in handling out-of-distribution HTML and compositional tasks. On the MiniWoB benchmark, we improve previous approaches using only HTML input by more than 17.7%, even surpassing the performance of RL-finetuned models. On the recent WebShop benchmark, our 3-billion-parameter model achieves superior performance to the existing state-of-the-art PaLM-540B. We also collect 347K gold demonstrations using our trained models, 29 times larger than prior work, and make them available to promote future research in this area. We believe that our work is a step towards building capable and generalist decision making agents for computer interface.

## 1 INTRODUCTION

Foundation models (Bommasani et al., 2021), especially large language models (LLM) (Brown et al., 2020; Chowdhery et al., 2022), have demonstrated incredible performance in commonsense, symbolic, arithmetic, and multi-step logical reasoning (Wei et al., 2022b;c; Kojima et al., 2022). Many prior works have shown that these models are capable of solving wide ranges of interactive decision making problems in the wild, much like generalist agents, including task planning in robotics (Huang et al., 2022a;b; Shah et al., 2022; Ahn et al., 2022), board game (Meta Fundamental AI Research Diplomacy Team et al., 2022), web-based retrieval and browser crawling (Nakano et al., 2021; Gur et al., 2022; Yao et al., 2022b; Zaheer et al., 2022).

Despite significant successes, existing LLM-based agents are only able to perceive their environments via text inputs (Nakano et al., 2021; Gur et al., 2022; Yao et al., 2022b). Even in robotics where visual perception is essential for decision making, scene perceptions are entrusted to object recognition modules (Gu et al., 2021b; Kamath et al., 2021) and described in a text format with fixed prompts (Zeng et al., 2022; Ahn et al., 2022; Huang et al., 2022b). The need to encode environment observations exclusively as text can limit the capability of spatial understanding for multi-step reasoning problems. For instance, in our daily lives, we humans use computers or crawl browsers by not only reading the contents of webpages, but also by recognizing the visual elements on the screen and their arrangement. In order to handle complex decision making tasks, it is necessary to ground text understanding and visual perception.

In this paper, we propose *Web navigation via Grounded Understanding Models* (WebGUM), a foundation model finetuned with a large corpus of multimodal web data to obtain a grounded vision-and-HTML understanding for autonomous web navigation (Shi et al., 2017; Liu et al., 2018; Gur

---

[*]Work done as Student Researcher at Google.

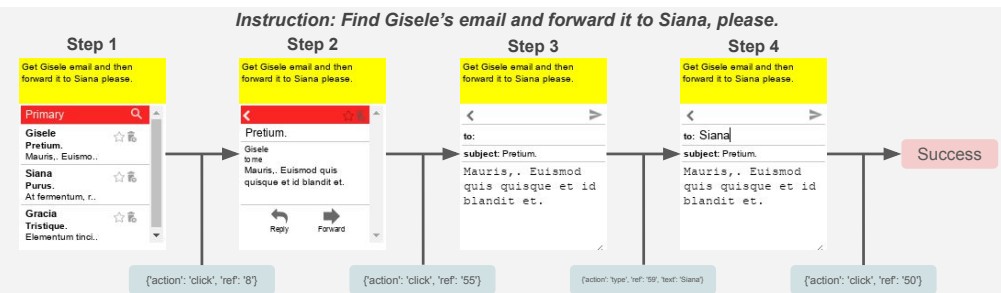

Figure 1: Example episode on MiniWoB++ (Shi et al., 2017; Liu et al., 2018) (`email-inbox-forward-nl`). The agent clicks the email from the proper sender, and types the correct receiver to forward that email, to satisfy the given instruction (e.g. *Find Gisele's email and forward it to Siana, please*). WebGUM makes use of both HTML and image screenshot information to adapt a pretrained instruction-finetuned foundation model to solve challenging web-based tasks such as this one.

et al., 2019). As shown in Figure 1, our model takes in a command for a web-based task via a natural language instruction (e.g., in an email client, *Find Gisele's email and forward it to Siana, please.*) and uses multimodal observations of the computer interface to complete the task via a sequence of computer actions such as *click* and *type*. We embed HTML and screenshot of the websites into shared multimodal tokens for spatial and semantic understanding of the scene. Moreover, to enhance the alignment with the user's intention for task accomplishment, we leverage an instruction-finetuned LLM (Wei et al., 2022a; Chung et al., 2022; Ouyang et al., 2022; Iyer et al., 2022) instead of unsupervised text-to-text pre-trained LLMs (Raffel et al., 2020; Brown et al., 2020) advocated by previous work (Gur et al., 2022). Through evaluation on MiniWoB++ (Shi et al., 2017; Liu et al., 2018), a representative web navigation benchmark with simulated websites, our multimodal model outperforms previous finetuned-LLM approaches trained with HTML inputs (Gur et al., 2022) by 17.7%. Our proposed WebGUM also surpasses existing approaches using reinforcement learning (RL) (Liu et al., 2018). We find that our models are especially adept at handling unknown composition of the tasks or out-of-distribution HTML inputs, synthesized with realistic perturbations.

Our extensive and precise ablations reveal the benefit of each of our contributions towards WebGUM's final performance; namely, the use of (1) multimodal vision-and-HTML observations, (2) instruction-finetuned language models, and (3) massive expert demonstrations. WebGUM could leverage multimodal tokens to ground vision and HTML understanding on the computer interface, especially to solve the multi-step reasoning tasks or the tasks that require global contexts, such as browser-crawling or dropdown calendar. Besides, we find that instruction-finetuned language models (Chung et al., 2022) remarkably boost the web navigation performance; compared to unsupervised pre-trained models (Raffel et al., 2020), it improves the success rate on MiniWoB++ by over 10%. On the recent WebShop (Yao et al., 2022a) benchmark, WebGUM also achieve superior performance to the existing state-of-the-art PaLM-540B (Yao et al., 2022b; Chowdhery et al., 2022), while our model only has 3 billion parameters. To be best of our knowledge, we are the first to demonstrate that instruction-finetuned LLM plays a critical role even in interactive decision making as well as common NLP tasks, and can transfer their notable performances to multimodal settings. Finally, we collect 347K multimodal expert demonstrations on MiniWoB++ with finetuned-LLM and scripted policy, 29 times larger than existing unimodal dataset (Liu et al., 2018), and make these publicly available for future research [1]. Our results also imply the scaling effects in web navigation; the model performance gradually increases as the dataset or model size does.

## 2 RELATED WORK

**Web Navigation** Autonomous web navigation is a sequential decision making problem where the agent controls computers or crawls the Internet on the browser to satisfy given instructions (Shi et al., 2017), such as form-filling (Diaz et al., 2013), information retrieval, or question answering (Nogueira

---

[1] https://github.com/google-research/google-research/tree/master/mm_webnav

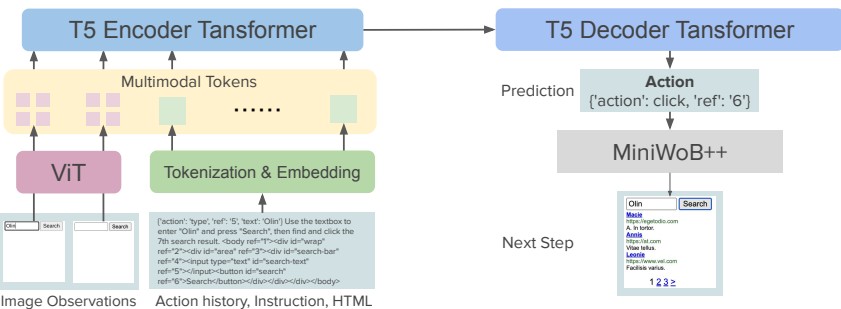

Figure 2: Overview of WebGUM, our multimodal encoder-decoder transformer model. It takes recent $H$-step screenshots ($H = 2$), action history, instruction, and HTML as inputs. Image observations are embedded to tokens per $16 \times 16$-size patch via pre-trained vision transformer (ViT) (Dosovitskiy et al., 2020)). Multimodal language-image tokens are fed into pre-trained T5 encoder-decoder transformer (Raffel et al., 2020), and then predict executable actions in text formats.

& Cho, 2016; Adolphs et al., 2022), which seems to be an important application for artificial intelligence to assist our daily lives (Mazumder & Riva, 2020; Li et al., 2020; Shvo et al., 2021).

While many kinds of benchmarks have been proposed (Toyama et al., 2021; Burns et al., 2022; Yao et al., 2022a), the most inclusive and representative benchmark to test the capability of autonomous agents is MiniWoB++ (Shi et al., 2017; Liu et al., 2018) because it consists of a set of simulated websites with various user instructions from primitive tasks to complex multi-step decision making tasks, such as sending emails or booking flights. Prior works have tried to solve this benchmark using a variety of techniques, including (1) RL with high-level workflow guidance (Liu et al., 2018) or with curriculum learning (Gur et al., 2019; 2021), (2) behavioral cloning and RL-finetuning with a large million-scale unreleased dataset (Humphreys et al., 2022), and (3) finetuned-LLM (Gur et al., 2022). Many of these approaches depend on specific structural bias based on the document object model (DOM) (Jia et al., 2019; He et al., 2020), or result in relatively lower performance if lacking tremendous labeled online (i.e., RL) interaction (Humphreys et al., 2022), which is difficult to collect from real websites as there is typically no reward signal and interactions are costly. In contrast, we ground the understanding of vision and HTML to solve canonical web-based tasks, and leverage the capability of instruction-finetuned LLM for strong inductive bias on multi-step reasoning and alignment with user intentions, while eschewing any online web interactions.

**Document Understanding** Several works have tackled document understanding with (multimodal) transformer models (Xu et al., 2019; Li et al., 2021a;c; Appalaraju et al., 2021; Tang et al., 2022; Wang et al., 2022a;b), including markup languages such as HTML (Aghajanyan et al., 2021; 2022; Li et al., 2021b; Lee et al., 2022a) for summarization of the documents or question answering on the contents. Despite the great efforts on document understanding, these works are less connected to interactive decision making problems. Our model obtains not only a grounded understanding of websites in a multimodal manner but also the ability to decide the optimal actions to achieve given instructions in web navigation, helping multi-step reasoning and global context perception.

**Multimodal Large-scale Models** Large language models have shown us incredible emergent abilities on a variety of NLP tasks, such as commonsense question answering, arithmetics, logical reasoning, open-ended text generation (Radford et al., 2019; Brown et al., 2020; Chowdhery et al., 2022; Wei et al., 2022b; Tay et al., 2022), or code completion (Chen et al., 2021b; Austin et al., 2021; Li et al., 2022b). In addition, some works have investigated vision-and-language understanding to improve the accuracy of common vision-based tasks such as open-ended image/object classification (Radford et al., 2021; Gu et al., 2021b; Kamath et al., 2021), image captioning, or visual question answering (Lu et al., 2022; Alayrac et al., 2022; Chen et al., 2022; Reed et al., 2022). Meanwhile, we focus on grounding the contents of visual and HTML inputs in instruction-finetuned LLM with a posteriori finetuning for autonomous web navigation.

**Foundation Models for Decision Making** In sequential decision making problems, such as task planning in robotics (Ahn et al., 2022; Huang et al., 2022a;b; Zeng et al., 2022), information retrieval (Yao et al., 2022b), or board game (Meta Fundamental AI Research Diplomacy Team et al., 2022), the ability of multi-step reasoning and strong inductive bias in foundation models are leveraged to solve complex tasks with few-shot in-context examples. Even in continuous control (Chen et al.,

| Methods | Training | Modality | Pre-trained Models | Dataset | Success Rate |
|---|---|---|---|---|---|
| CC-Net (Humphreys et al., 2022) | SL | DOM+Image | ResNet | 2.4M | 32.0% |
| WebN-T5 (Gur et al., 2022) | SL | HTML | T5-XL | 12K | 48.4% |
| WGE (Liu et al., 2018) | SL+RL | DOM | – | 12K+ | 64.6% |
| CC-Net (Humphreys et al., 2022) | SL+RL | DOM+Image | ResNet | 2.4M+ | 96.4% |
| WebGUM (Ours) | SL | HTML | Flan-T5-XL | 347K | 61.5% |
| WebGUM (Ours) | SL | HTML+Image | Flan-T5-XL, ViT-B16 | 347K | 66.1% |

Table 1: Average success rate on MiniWoB++ among 56 tasks. We recalculate the baseline performances referring Humphreys et al. (2022) and Gur et al. (2022). See Appendix D for the detailed scores per task. WebGUM significantly outperforms previous finetuned-LLM approach (Gur et al., 2022) which is state-of-the-art among methods trained with supervised learning (SL). When comparing to existing methods that leverage online reinforcement learning (SL+RL), our proposed WebGUM exceeds the baseline from Liu et al. (2018). Despite the superior performance, our SL model is still behind SL+RL state-of-the-art (Humphreys et al., 2022) due to the data coverage in the training dataset and lack of exploration during RL-finetuning. "+" in Dataset column means that the number of episodes, required during RL training steps, is not included because no details were described in their works. Videos are available at https://sites.google.com/view/mm-webnav/.

2021a; Janner et al., 2021; Furuta et al., 2022b; Brohan et al., 2022) or computer games (Reed et al., 2022; Lee et al., 2022b; Fan et al., 2022), high-capacity transformer models are trained with a large amount of diverse dataset via multi-task behavioral distillation (Chen et al., 2021c; Gu et al., 2021a; DeepMind Interactive Agents Team et al., 2021; Furuta et al., 2022a; Shridhar et al., 2022; Jiang et al., 2022). To build autonomous web navigation agents, we also leverage pre-trained LLM (Raffel et al., 2020; Chung et al., 2022), finetuned with massively-curated multimodal demonstrations, and to be best of our knowledge, we are the first to demonstrate that instruction-finetuned LLM (Chung et al., 2022) is essential for the notable performance on interactive decision making in addition to common NLP tasks.

## 3 PRELIMINARIES

We formulate autonomous web navigation as a deterministic sequential decision making problem; composed of a state space $\mathcal{S}$, action space $\mathcal{A}$, deterministic transition function $T : \mathcal{S} \times \mathcal{A} \to \mathcal{S}$, instruction space $\mathcal{G}$, reward function (or episodic success criteria) $r : \mathcal{S} \times \mathcal{G} \times \mathcal{A} \to \{0, 1\}$. At each time step $t$, the agent follows a parameterized policy conditioned on previous states and actions $\pi : \underbrace{\mathcal{S} \times \cdots \times \mathcal{S}}_{\times t} \times \underbrace{\mathcal{A} \times \cdots \times \mathcal{A}}_{\times t} \times \mathcal{G} \to \mathcal{A}$, and transits to the next state: $s_{t+1} = T(s_t, a_t)$. This process continues until the agent reaches the terminal state (e.g. Submit button is clicked) or the max time step is exceeded. The episode is treated as a success if given instruction $g$ is satisfied (i.e. $r(s_t, g, a_t) = 1$), and as a failure if the agent takes a invalid action or reaches a wrong terminal state.

In autonomous web navigation, the state $s_t \in \mathcal{S}$ is a web page consisting of the raw HTML as a text sequence and a screenshot as an image. Following prior works (Shi et al., 2017; Liu et al., 2018; Gur et al., 2019; 2021), we assume the constraint action space: function(selector, text). function is either *click* or *type*, selector is an integer index that can uniquely specify the element, and text is a text input for *type* function.

Figure 1 presents one of the example episodes on MiniWoB++ (Shi et al., 2017; Liu et al., 2018). To meet the given instruction, the agent clicks an email from the proper sender and types the correct receiver to forward that email. MiniWoB++ includes such multi-step decision making tasks, as well as primitive behavioral tasks; for instance, clicking buttons or entering texts. Past work has proposed to solve MiniWoB++ tasks using supervised-learned (SL) agents trained with expert demonstrations (Humphreys et al., 2022; Gur et al., 2022), reinforcement-learned (RL) agents with specialized neural network architectures (Jia et al., 2019; Gur et al., 2019), as well as agents trained with SL plus RL-finetuning (Liu et al., 2018; Humphreys et al., 2022).

## 4 WEBGUM

### 4.1 MULTIMODAL TRANSFORMER MODELS

We extend the encoder-decoder transformer (Vaswani et al., 2017), proposed in Raffel et al. (2020) to the multimodal model as shown in Figure 2. The model is fed with visual tokens embedded from historical image observations ($H = 2$), and text tokens from action history, user instruction, and raw HTML. Encoder transformer handles both visual and text tokens in a unified manner and then, decoder predicts text-format actions. Similar to Gur et al. (2022), we focus on encoder-decoder architectures to solve HTML-based web navigation tasks, because their bi-directional nature could leverage the tree structure of HTML and they scale better than other models. See Appendix A for further details.

**Image Encoder for Visual Tokens** We adopt vision transformer (ViT) (Dosovitskiy et al., 2020), pre-trained on the image classification task with ImageNet-21K (Deng et al., 2009), as an encoder to embed images into the visual tokens. To better extract spatial and semantic information from the screenshots of websites, we use the tokens per patch rather than the token per image (i.e. CLS-token). We divide an input image into $16 \times 16$ patches – giving a total of $14 \times 14$ (number of patches) $\times$ 2 (context window) $= 392$ visual tokens. We crop the screenshots of MiniWoB++ to remove the yellow instruction part (as shown in Figure 1), and the image size becomes $160 \times 160$. We pad cropped images with white pixels to fit them into $224 \times 224$; the input size for ViT.

### 4.2 INSTRUCTION-FINETUNED LARGE LANGUAGE MODELS

Since pre-trained LLM has strong reasoning abilities and inductive bias that should be applicable to any kind of NLP task (Raffel et al., 2020; Brown et al., 2020; Chowdhery et al., 2022; Wei et al., 2022b) and even to understanding HTML (Gur et al., 2022), we finetune pre-trained LLMs with a massive behavioral dataset on web navigation. Furthermore, we leverage Flan-T5 (Chung et al., 2022), an instruction-finetuned LLM, finetuned with large-scale instructions and few/zero-shot chain-of-thought examples, to enhance the alignment with the user's intention for task accomplishment, rather than unsupervised text-to-text pre-trained LLM (Raffel et al., 2020) used in relevant work (Gur et al., 2022). Note that the training dataset for Flan-T5 contains programming language corpus and code completion tasks (in Muffin), while one for original T5 does not.

Since instruction-finetuned LLM presents drastic improvements on many common NLP tasks (Ouyang et al., 2022; Chung et al., 2022; Iyer et al., 2022), we could expect the performance improvements even in interactive decision making problems. We mainly adopt the XL-size model, which shows enough and great capability for reasoning with about 3 billion parameters.

### 4.3 MASSIVE DATASET COLLECTION WITH FINETUNED LLM

Recent successes of foundation models are largely powered by internet-scale data (Brown et al., 2020; Radford et al., 2021; Chen et al., 2022; Wang et al., 2023). While large amount of data is critical, for web navigation domain, there is only a small public dataset for MiniWoB++, consisting of 12K episodes of human demonstration (Liu et al., 2018). Moreover, the dataset only consists of DOM observations and lacks any visual features, which might limit the spatial perception of the elements on the page. A large-scale multimodal dataset, including screenshots of websites, is required to build a better navigation policy at scale.

To collect a huge amount of multimodal behavioral dataset on MiniWoB++, we leverage a public finetuned-LLM policy (Gur et al., 2022) trained with multi-task human demonstration dataset (Liu et al., 2018) for data collection instead of hiring human demonstrators, which significantly reduces the cost to construct a new dataset by leveraging the prior success of autonomous agents. We gradually increase the dataset size; we first rollout a LLM policy with 100 episodes per task, and only keep the successful trajectories, which results in a 2.8K-episode dataset. Then, we train other models with this dataset and use them for data collection again. We run those models with 10,000 episodes per task and discard failure cases. In addition, to collect expert demonstrations on a harder task that finetuned-LLM struggles to solve, we write a scripted policy for `book-flight` task. Such efforts result in a multi-task 347K-episode dataset with HTML and screenshots at each time step, generated by proficient autonomous agents. See Appendix B for further details.

| Methods | Modality | Success Rate |
|---|---|---|
| WebGUM | HTML | 61.5% |
| WebGUM (white) | HTML+Image | 61.5% |
| WebGUM (random) | HTML+Image | 62.2% |
| WebGUM | HTML+Image | 66.1% |
| WebGUM (single, $H = 2$) | HTML+Image | 63.6% |
| WebGUM (multiple, $H = 1$) | HTML+Image | 64.8% |
| WebGUM (multiple, $H = 2$) | HTML+Image | 66.1% |

Table 2: Average success rate with white/random image inputs, single/multiple visual tokens, and context length ($H = 1, 2$). All models are initialized with Flan-T5-XL and ViT-B16, and trained with our 347K-episode dataset. The results imply that WebGUM successfully leverages semantic and spatial information from image modality, and multiple visual tokens from patches could extract much richer features than a single visual token per image.

## 5 RESULTS

We test our method on the MiniWoB++ benchmark (Shi et al., 2017; Liu et al., 2018) with 100 evaluation episodes per task, taking the average success rate over 56 tasks taken from Gur et al. (2022). Due to the huge computational requirements, we run one seed to train each model throughout the paper. Table 1 shows that our proposed WebGUM, especially multimodal model, significantly outperforms the previous best SL model (Gur et al., 2022) over 17.7% and exceeds WGE (Liu et al., 2018), an RL-finetuned baseline (average of single-task models), over 1.7%[2]. Despite the superior performance of our SL model, we are still behind SL+RL-finetuned state-of-the-art (Humphreys et al., 2022) due to the data coverage in the training dataset and lack of exploration during RL-finetuning. However, compared to its SL-only model, our method achieves double the performance even with a 7 times smaller dataset, which may reduce the required episodes for RL-finetuning and can bridge the gap between SL and RL as better behavioral priors. We believe improving SL models is a valuable contribution as a scalable and deployable approach towards real-world web automation where online interactions are costly.

In the following sections, we do extensive and precise ablations of our design choices for WebGUM presented in Section 4: image modality (Section 5.1), instruction-finetuned LLM (Section 5.2) and its application in the recent WebShop (Yao et al., 2022a) benchmark (Section 5.3). We also investigate the effect of dataset and model size on task success (Section 5.4). Furthermore, we examine the robustness and generalization of WebGUM with realistic input corruptions and unknown compositions of the tasks (Section 5.5).

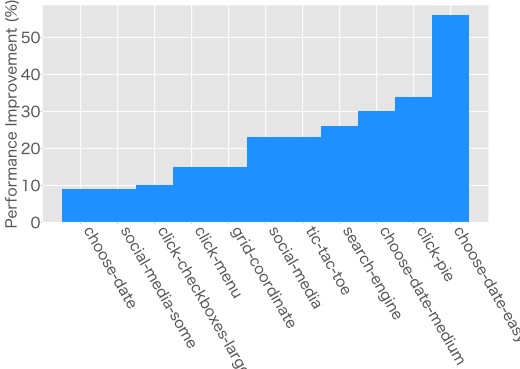

Figure 3: Top-10 performance improvement by adding image modality to HTML on 56 tasks from MiniWoB++. We subtract the success rates to compute absolute improvement: (Success Rate of WebGUM(HTML+Image)) - (Success Rate of WebGUM(HTML)). Image modality seems to be leveraged for multi-step reasoning tasks with page transitions or tasks that require global contexts (e.g. tic-tac-toe or grid-coordinate) See Appendix D and G for the details.

---

[2]Videos are available at https://sites.google.com/view/mm-webnav/

| Pre-Trained Models | Modality | Success Rate |
|---|---|---|
| T5-XL (Gur et al., 2022) | HTML | 48.4% |
| T5-XL, ViT-B16 | HTML+Image | 55.6% |
| Flan-T5-XL | HTML | 61.5% |
| Flan-T5-XL, ViT-B16 | HTML+Image | 66.1% |

Table 3: Average success rate with different pre-trained models. We refer Gur et al. (2022) for T5-XL result and other models are trained with our 347K-episode dataset. In both modalities, instruction-finetuned LLM checkpoints (Flan-T5) outperform unsupervised LLM checkpoints (T5) by a large margin (over 10%).

## 5.1 Does Image Modality Help for Task Success?

To examine if the models actually leverage image modality for a grounded understanding of websites, we design two ablations: replacing image observations with completely white images, and with randomly sampled MiniWoB++ screenshots taken in the initial states. In addition, we also investigate whether our design choices for image observations (multiple tokens from patches, historical observations with $H = 2$) are suitable ones or not.

Table 2 reveals that the performance of the model with white images, is comparable to the unimodal HTML model. Because the model with randomly-taken images may accidentally contain the images from the same task to solve, WebGUM (random) slightly surpasses WebGUM (white). These results prove WebGUM successfully obtains grounded vision and HTML understanding for web navigation by leveraging semantic and spatial information from image modality. Multiple visual tokens from patches outperform a single visual token per image, which means they extract much richer task-relevant features. Besides, we find that historical image observations ($H = 2$) contribute to the improvement more than single-step observation ($H = 1$).

We also compare per-task performance gaps caused by adding image modality to instruction-finetuned LLM. Figure 3 presents the top-10 absolute performance improvement, which suggests WebGUM leverages visual inputs for multi-step reasoning tasks with page transitions (e.g. `choose-date-easy` or `-medium`) or the tasks that require global context perception of the page (e.g. `tic-tac-toe` or `grid-coordinate`). See Appendix D and G for further details.

| Methods | Training | Models | Score | Success Rate |
|---|---|---|---|---|
| Rule | – | – | 45.6 | 9.6% |
| IL | SL | BART, BERT | 59.9 | 29.1% |
| IL+RL | SL+RL | BART, BERT | 62.4 | 28.7% |
| Act | In-context | PaLM-540B | 62.3 | 30.1% |
| ReAct | In-context | PaLM-540B | 66.6 | 40.0% |
| WebGUM | SL | Flan-T5-XL | **67.5** | **45.0%** |

Table 4: Average score and success rate on WebShop. WebGUM achieves 45.0% success, outperforming baseline approaches including ReAct, a prompted PaLM-540B. We refer Yao et al. (2022b) for the baselines.

## 5.2 Do Instruction-Finetuned Language Models Help for Task Success?

Because web navigation problem is at the intersection of RL, NLP, and vision-and-language domains, one natural question is whether we could leverage the progress in other domains for sequential decision making. Following the success in many NLP tasks (Ouyang et al., 2022; Iyer et al., 2022), we test instruction-finetuned LLM (Chung et al., 2022) as a pre-trained model for web navigation policy, compared to unsupervised LLM (Raffel et al., 2020) used in prior work (Gur et al., 2022).

Table 3 first shows that image modality also improves the performance of T5-initialized multimodal models (+7.2%) as the same as Flan-T5-initialized models. Despite such performance gain, Table 3 proves that instruction-finetuned LLM checkpoints, Flan-T5, improves the success rate compared to unsupervised LLM checkpoints, original T5, by a large margin (+13.1% in HTML and +10.5% in HTML+image model). To be best of our knowledge, these results are the first to demonstrate that instruction-finetuned LLMs are beneficial even for interactive decision making as well as common NLP tasks, and can transfer their notable performances to multimodal settings. These facts must be

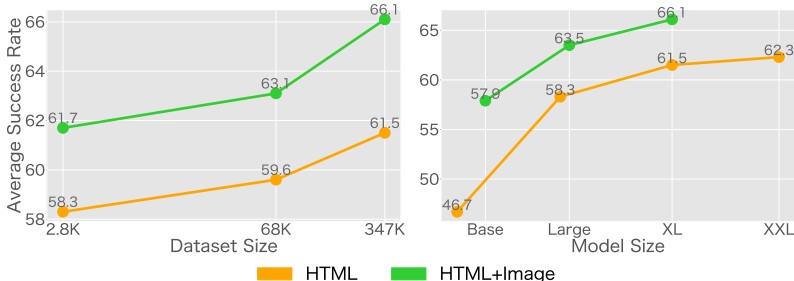

Figure 4: Average success rate of WebGUM with different dataset (left) and model sizes (right). X-axis is a logarithmic scale. As for both HTML and multimodal models, we could observe the scaling effect: the larger the dataset and model size are, the higher the success rates are. Surprisingly, our approach outperforms previous SL state-of-the-art (48.4% by Gur et al. (2022)) more than 9.9% even with 2.8K-episode dataset (about 25% of the previous dataset curated by Liu et al. (2018)). See Appendix C for further details.

preferable, because we could expect to leverage the innovations on NLP to tackle complex decision making problems.

## 5.3 DO INSTRUCTION-FINETUNED LANGUAGE MODELS ALSO WORK ON WEBSHOP?

We extensively evaluate our WebGUM on WebShop (Yao et al., 2022a), an online-shopping website simulator with a large amount of real-world product data. Because it requires complex multi-step reasoning considering previous contexts for comparison, WebShop is suitable for investigating the capability of instruction-finetuned LLM in decision making tasks in depth. WebShop provides a user instruction that describes the features of item (e.g. *I need a long clip-in hair extension which is natural looking, and price lower than 20.00 dollars*). The agents should search, compare and choose a proper product that matches the given instruction. The performance score is evaluated by the percentage of required attributes covered by the chosen product, and if the product meets all the requirements, that episode is labeled a success. See Appendix F for further details.

Table 4 shows that WebGUM achieves 45.0% success, significantly outperforming not only simple baselines, such as supervised imitation learning (IL) and IL plus RL-finetuing (by more than 15%), but also recent prompt-based LLM agents, including ReAct (Yao et al., 2022b) (i.e. PaLM-540B (Chowdhery et al., 2022) with one-shot prompt and reasoning annotations), while our model only has 3 billion parameters. Due to the consistent reasoning and enhanced alignment with user's intentions in instruction-finetuned LLMs, WebGUM could compare the products with backtracking, and choose proper options (see Appendix G).

## 5.4 SCALING EFFECT IN DATASET AND MODEL SIZE

Large-scale models often show their incredible capability by scaling tremendous data size and high-capacity model size (Shoeybi et al., 2019; Brown et al., 2020; Kaplan et al., 2020; Rae et al., 2021; Radford et al., 2021; Wei et al., 2022b; Chowdhery et al., 2022). We investigate whether similar scaling effects might be observed in web navigation by increasing the number of episodes for training, and the number of parameters for the transformer architectures.

To investigate the scalability to the dataset size, we prepare three dataset: minimal 2.8K demonstrations, 347K demonstrations, and its 20%-size demonstrations (68K). Figure 4 (left) proves that increasing dataset size leads to the improvement of the average success rate. Notably, WebGUM with only 2.8K HTML episodes already achieves 58.3%, outperforming previous SL state-of-the-art (48.4% by Gur et al. (2022)) more than 9.9%; that dataset size is about 25% of the previous dataset released by Liu et al. (2018). This surprising data-efficiency might come from the sufficient inductive bias and alignment with the user intentions in instruction-finetuned LLMs, and our approach could fully leverage them for sequential web automation problems.

In addition to dataset size, Figure 4 (right) shows that the performance of WebGUM improves as the number of parameters in T5 model increases from Base (220M) to XXL (11B). These results would encourage the community to pay more attention to the enlargement of the dataset and model capacity

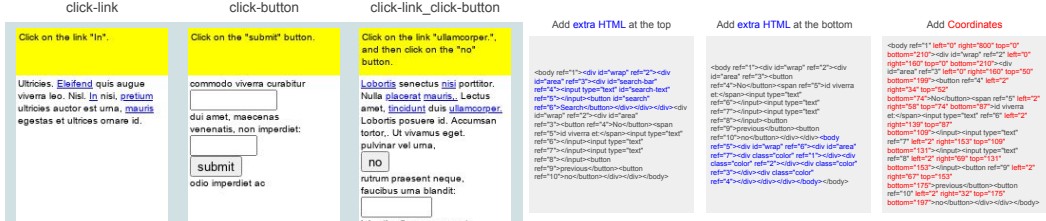

Figure 5: **(Left)** Example of compositional evaluation on MiniWoB++. We combine two different tasks (`click-link` and `click-button`) into a single-page sequential task (`click-link_click-button`). See Appendix E for the details of combinations. **(Right)** Example of input perturbation for MiniWoB++ evaluation. We prepare three different types of perturbations at test time: adding extra HTML at the top of the original input HTML (left) or at the bottom of HTML (middle), and adding task-irrelevant attributes such as coordinate information (right). We randomly sample extra HTML from the human-collected 12K dataset (Liu et al., 2018). This example HTML is taken from `click-button`.

for decision making agents as implied in data-driven prior works (Lee et al., 2022b; Brohan et al., 2022; Furuta et al., 2022a; Fan et al., 2022; Reed et al., 2022; Jiang et al., 2022). See Appendix C for further details.

### 5.5 DOES WEBGUM GENERALIZE TO REALISTIC COMPOSITIONAL TASKS OR INPUT PERTURBATIONS?

Generalization to the out-of-distribution inputs or unseen combination of known tasks are important challenges for the web navigation agents to be deployed on the real-world Internet, but have often been missed in previous works. To investigate the generalization capability of our proposed methods, we test (1) generalization to the compositional tasks, and (2) robustness to the input perturbations.

For the compositional tasks, we pick up 4 `click-`"something" (link, button, checkboxes, dialog) tasks and make 6 combinations of these by naively stitching with 2 or 3 tasks (e.g. Figure 5). These tasks should be resolved in order. See Appendix E for further details. Table 5 shows that WebGUM with HTML and image inputs outperforms prior finetuned-LLM by over 12.5%, which implies WebGUM has obtained better primitive skills to control computers and could transfer them to resolve unseen tasks.

To check the robustness against input corruptions, we test three different realistic perturbations; adding extra HTML at the top or bottom of the original HTML, and adding attributes of coordinates (left, right, top, bottom) in each element of HTML at test time. These perturbations often happen in the real world due to the renewal or API changes, not to mention unknown websites, and rule-based pre-processing may not fully cover them. Table 6 shows that while all the methods are affected by the input corruptions to some extent, WebGUM, with both HTML and HTML plus image modalities, achieves significantly better performances than Gur et al. (2022). Notably, our multimodal WebGUM significantly outperforms prior finetuned-LLM (+ 33.9%) and unimodal HTML model (+11.7%) when extra attributes of coordinate to HTML are added, which also supports the fact that WebGUM leverages semantic information extracted from visual tokens.

| Methods | Modality | Success Rate |
|---|---|---|
| WebN-T5 (Gur et al., 2022) | HTML | 51.0% |
| WebGUM | HTML | 61.7% |
| WebGUM | HTML+Image | **63.5%** |

Table 5: Average success rate on 6 compositional MiniWoB tasks. WebGUM generalizes combinational tasks better than Gur et al. (2022), achieving better success rate by over 12.5% (HTML+Image) or 10.7% (HTML).

### 6 DISCUSSION AND LIMITATION

Throughout the paper, we present an effective and practical methodology to distill the multi-task, multimodal behavioral data into instruction-finetuned LLMs via supervised finetuning. We leave

| Methods | Modality | Perturbation | Success Rate |
|---------|----------|--------------|--------------|
| WebN-T5 (Gur et al., 2022) | HTML | Top
Bottom
Coordinates | 24.7%
42.8%
6.4% |
| WebGUM | HTML | Top
Bottom
Coordinates | 34.8%
46.4%
28.6% |
| WebGUM | HTML+Image | Top
Bottom
Coordinates | **37.7%**
**49.1%**
**40.3%** |

Table 6: Average success rate of perturbation evaluation on MiniWoB++, 56 tasks. We test three different perturbation evaluations; adding extra HTML at the top/bottom of the original HTML, and adding attributes of coordinates (left, right, top, bottom) in each element of HTML at test time. The results show that while all the methods are affected by input corruptions to some extent, our WebGUM, especially multimodal model, achieves significantly better performances than previous finetuned-LLM.

finetuning large-scale multimodal transformers with RL (Liu et al., 2018; Jaques et al., 2019; Ziegler et al., 2019; Stiennon et al., 2020; Nakano et al., 2021; Ouyang et al., 2022; Humphreys et al., 2022) in a scalable manner as future work, which is a powerful tool for output alignment with user intentions or preferences. We collect and release a multimodal expert dataset with 347K episodes on MiniWoB++. However, this is still far from internet-scale dataset that is necessary for generalist models. Collecting behavioral data at scale by iterative data-collection and deployment (Ghosh et al., 2021; Matsushima et al., 2021; Li et al., 2022a) might be a key for practical interactive agents.

Since our approach – taking raw HTML and screenshots as inputs and predicting executable actions directly – has minimal assumptions that constraint model architectures, it might be applicable to a wide range of computer tasks. More flexible action space, such as pixel-level clicking, scrolling page, or dragging elements would lead to much better generalization. While we show that WebGUM could deal with compositional and perturbed tasks in a robust way, human-level broader generalization to the diverse real-world websites is still a hard problem to be resolved.

## 7    CONCLUSION

To ground vision-and-HTML understanding for web-based sequential decision making problems, we develop *Web navigation via Grounded Understanding Models* (WebGUM) by finetuning an instruction-finetuned foundation model with multimodal and proficient demonstrations in web navigation. WebGUM significantly improves the success rate on MiniWoB, compared to previous finetuned-LLM baseline from 48.4% to 66.1%, deals with out-of-distribution HTML and unseen compositional tasks much better, and achieves better performance than PaLM-540B in WebShop. Our detailed ablations reveal that (1) multiple visual tokens extract spatial and semantic information to aid the multi-step reasoning and global context perception, and (2) instruction-finetuned language models remarkably boost web navigation performance due to the better alignment with user instructions and the transferability to multimodal settings. Furthermore, we publicly release 347K multimodal expert demonstrations on MiniWoB++, which is about 29 times larger than the existing dataset. We hope our work would inspire the community to build more capable and general decision making models for autonomous web navigation.

## ACKNOWLEDGEMENTS

HF was supported by JSPS KAKENHI Grant Number JP22J21582. We thank Yusuke Iwasawa, Mustafa Safdari, Austin Huang for helpful feedback on this work, and Shunyu Yao for setting up WebShop experiments.

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

APPENDIX

# A    IMPLEMENTATION DETAILS

We adopt the encoder-decoder models proposed by Raffel et al. (2020) as multimodal transformers, and vision transformer (Dosovitskiy et al., 2020) pre-trained with ImageNet-21K (Deng et al., 2009) as an image encoder for the visual tokens[3]. We especially use ViT-B16, a small-size transformer with 86 million parameters, which divides an input image into $16 \times 16$-size patches. We use publicly available checkpoints of T5 (Raffel et al., 2020)[4], Flan-T5 (Chung et al., 2022)[5], and T5-XL finetuned with MiniWoB++ demonstrations (Gur et al., 2022)[6] for the experiments. As suggested in Gur et al. (2022), we focus on encoder-decoder architectures to solve HTML-based web navigation tasks. Applying our method to other architectures, such as auto-regressive decoder-only models (Radford et al., 2019; Brown et al., 2020; Chowdhery et al., 2022) remains as future work. To construct the training pipeline, we leverage SeqIO (Roberts et al., 2022) library, and use SentencePiece (Kudo & Richardson, 2018) vocabulary with 32K tokens from C4 dataset (Raffel et al., 2020) for text tokenization. The batch size for training is 128 (256 for XXL-size model), and input sequence length is set to 512.

## A.1    SHORTEN HTML INPUT WHILE PRESERVING STRUCTURAL BIAS

As a markup language, HTML strongly holds the structural information, but it often contains task-irrelevant, seemingly redundant parts as inputs for language models, which may affect the performance. If we effectively shorten HTML while still keeping the structural properties of markup programming language to some extent, that would be beneficial to solve the task. Motivated by this intuition, we remove closing tags (e.g. `</body>`) from the inputs of language models (Figure 6) at inference time.

We find this technique slightly improves the success rate on MiniWoB++; we test WebGUM and finetuned-LLM baseline, and use T5-XL checkpoint released by Gur et al. (2022) for comparison. Table 7 reveals that WebGUM consistently improves the performance in both HTML (+4.1%) and HTML+image modalities (+1.3%), while T5-XL, trained with human-collected 12K dataset (Liu et al., 2018), decreases performance (-4.6%). The results suggest that our WebGUM is robust to the changes in input format and can benefit from removing redundant parts of HTML. This might also be because Flan-T5 is finetuned with code completion tasks during an instruction-finetuning phase, while T5 training corpus does not include programming code.

---

[3]https://github.com/google-research/scenic
[4]https://github.com/google-research/t5x/blob/main/docs/models.md#t5-11-checkpoints
[5]https://github.com/google-research/t5x/blob/main/docs/models.md#flan-t5-checkpoints
[6]https://console.cloud.google.com/storage/browser/gresearch/webllm/webn_t5_3b

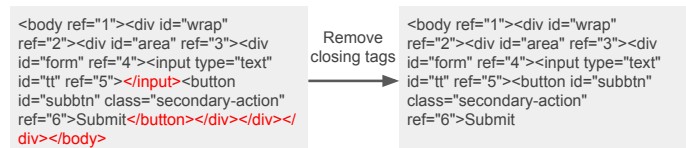

Figure 6: Example of shortening input HTML by removing closing tags (e.g. ``). Red part seems to be redundant to solve the tasks. HTML is taken from `enter-text`.

| Methods | Modality | Dataset | Success Rate |
|---|---|---|---|
| WebN-T5 (Gur et al., 2022) | HTML | 12K | 48.4% |
| WebN-T5 (w/o tags) | HTML | 12K | 43.8% |
| WebGUM (w/ tags) | HTML | 347K | 57.4% |
| WebGUM (w/o tags) | HTML | 347K | 61.5% |
| WebGUM (w/ tags) | HTML+Image | 347K | 64.8% |
| WebGUM (w/o tags) | HTML+Image | 347K | 66.1% |

Table 7: Average success rate with removing closing tags from HTML. The models are initialized with T5-XL or Flan-T5-XL (+ViT-B16) and trained with our 347K-episode dataset. While T5-XL, trained with human-collected 12K dataset (Liu et al., 2018), decreases performance, WebGUM consistently improves the performance in both HTML and HTML+image inputs. We also observed removing closing tags in HTML from the training dataset has a similar performance gain, but is slightly lower than only removing them at test time.

## B    DATASET DETAILS

To construct a large-scale multimodal behavioral dataset on MiniWoB++, we leverage a public finetuned-LLM policy (Gur et al., 2022) trained with multi-task human demonstration dataset (Liu et al., 2018)[7] as a demonstrator. We run LLM policies with 10,000 episodes per task and discard failure episodes. We also use a scripted policy for `book-flight` task, a harder task that finetuned-LLM policy cannot solve. Table 10 shows the details of our multimodal dataset, consisting of HTML, screenshots, actions, and instructions at each time step.

## C    DETAILS ON DATASET AND MODEL SIZE

We here test the different dataset and model sizes to reveal whether similar trends to NLP holds or not. As for both HTML and multimodal models, we could observe the scaling effects in web navigation: the larger the dataset (Table 8) and model (Table 9) size are, the higher the success rates are. Surprisingly, our approach with only 2.8K HTML episodes (about 25% of the previous dataset size curated by Liu et al. (2018)) already achieves 58.3%, outperforming previous SL state-of-the-art (48.4% by Gur et al. (2022)) more than 9.9%. Besides, instruction-finetuned models help Base-size to perform on par (46.7%) or outperform (57.9%) previous XL-size state-of-the-art (48.4%). This surprising efficiency might come from the sufficient inductive bias and alignment with the user intentions in instruction-finetuned LLMs, and our approach could fully leverage them for web automation problems. The margin of improvement might be smaller than expected due to the limited coverage of data collected by finetuned-LLM policies.

Table 8 also implies the quality of behaviors might be important, because WebGUM, initialized with Flan-T5-XL and trained with human-collected 12K dataset, is not so good as one trained with our 2.8K one. Since NLP tasks often only use correctly-annotated datasets for training, the dataset that contains hesitant or redundant behaviors might slightly hurt the performance of LLM-driven policies.

---

[7]https://github.com/stanfordnlp/miniwob-plusplus-demos

| Pre-Trained Models | Modality | Dataset | Success Rate |
|---|---|---|---|
| T5-XL (Gur et al., 2022) | HTML | 12K | 48.4% |
| Flan-T5-XL | HTML | 12K | 48.6% |
| Flan-T5-XL | HTML | 2.8K | 58.3% |
| Flan-T5-XL | HTML | 68K | 59.6% |
| Flan-T5-XL | HTML | 347K | 61.5% |
| Flan-T5-XL, ViT-B16 | HTML+Image | 2.8K | 61.7% |
| Flan-T5-XL, ViT-B16 | HTML+Image | 68K | 63.1% |
| Flan-T5-XL, ViT-B16 | HTML+Image | 347K | 66.1% |

Table 8: Average success rate of WebGUM with different dataset sizes. We observe the larger the dataset size is, the higher the success rate is. Surprisingly, our approach outperforms previous state-of-the-art by over 9.9% even with 2.8K-episode dataset (about 25% of the previous dataset curated by Liu et al. (2018)).

| Pre-Trained Models | # of Params | Modality | Success Rate |
|---|---|---|---|
| Flan-T5-Base | 220M | HTML | 46.7% |
| Flan-T5-Large | 770M | HTML | 58.3% |
| Flan-T5-XL | 3B | HTML | 61.5% |
| Flan-T5-XXL | 11B | HTML | 62.3% |
| Flan-T5-Base, ViT-B16 | 310M | HTML+Image | 57.9% |
| Flan-T5-Large, ViT-B16 | 860M | HTML+Image | 63.5% |
| Flan-T5-XL, ViT-B16 | 3B | HTML+Image | 66.1% |

Table 9: Average success rate of WebGUM with different model sizes. As for both HTML-only and multimodal models, we could observe the performance increases as the model size does.

| Task | # of episodes | # of steps | Ratio (episode) |
|---|---|---|---|
| book-flight | 9999 | 90177 | 2.88% |
| choose-date | 383 | 1508 | 0.11% |
| choose-date-easy | 3353 | 12946 | 0.97% |
| choose-date-medium | 2222 | 8733 | 0.64% |
| choose-list | 1861 | 3724 | 0.54% |
| click-button | 9782 | 9909 | 2.82% |
| click-button-sequence | 10000 | 20000 | 2.88% |
| click-checkboxes | 9761 | 28904 | 2.81% |
| click-checkboxes-large | 1962 | 19072 | 0.57% |
| click-checkboxes-soft | 9228 | 36384 | 2.66% |
| click-checkboxes-transfer | 10000 | 59793 | 2.88% |
| click-collapsible | 5947 | 13077 | 1.71% |
| click-collapsible-2 | 2199 | 5627 | 0.63% |
| click-color | 2554 | 2554 | 0.74% |
| click-dialog | 10000 | 10000 | 2.88% |
| click-dialog-2 | 3285 | 3285 | 0.95% |
| click-link | 9961 | 9961 | 2.87% |
| click-menu | 3238 | 3243 | 0.93% |
| click-option | 9998 | 20000 | 2.88% |
| click-pie | 3724 | 8548 | 1.07% |
| click-scroll-list | 0 | 0 | 0.00% |
| click-shades | 0 | 0 | 0.00% |
| click-shape | 6116 | 6117 | 1.76% |
| click-tab | 9978 | 13177 | 2.88% |
| click-tab-2 | 1844 | 2109 | 0.53% |
| click-tab-2-hard | 1574 | 1916 | 0.45% |
| click-test | 10000 | 10000 | 2.88% |
| click-test-2 | 10000 | 10000 | 2.88% |
| click-widget | 9963 | 9963 | 2.87% |
| count-shape | 5849 | 5893 | 1.69% |
| email-inbox | 5159 | 14258 | 1.49% |
| email-inbox-forward-nl | 9995 | 39980 | 2.88% |
| email-inbox-forward-nl-turk | 4900 | 20165 | 1.41% |
| email-inbox-nl-turk | 4346 | 11416 | 1.25% |
| enter-date | 10000 | 20000 | 2.88% |
| enter-password | 9980 | 29940 | 2.88% |
| enter-text | 10000 | 20000 | 2.88% |
| enter-text-dynamic | 9983 | 19966 | 2.88% |
| enter-time | 0 | 0 | 0.00% |
| focus-text | 10000 | 10000 | 2.88% |
| focus-text-2 | 10000 | 10000 | 2.88% |
| grid-coordinate | 8353 | 8353 | 2.41% |
| guess-number | 1021 | 2042 | 0.29% |
| identify-shape | 9007 | 9010 | 2.60% |
| login-user | 9793 | 29379 | 2.82% |
| login-user-popup | 9786 | 39170 | 2.82% |
| multi-layouts | 10000 | 40000 | 2.88% |
| multi-orderings | 10000 | 40000 | 2.88% |
| navigate-tree | 9864 | 15140 | 2.84% |
| search-engine | 8872 | 35095 | 2.56% |
| social-media | 2631 | 4407 | 0.76& |
| social-media-all | 95 | 208 | 0.03% |
| social-media-some | 319 | 893 | 0.09& |
| tic-tac-toe | 3947 | 13773 | 1.14% |
| use-autocomplete | 3465 | 6930 | 1.00% |
| use-spinner | 530 | 532 | 0.15% |
| **Total** | 346827 | 867277 | 100% |

Table 10: Details of our multimodal dataset. It contains about 347K episodes in total.

## D  PER-TASK PERFORMANCE OF MINIWOB++

In this section, we present per-task success rate on MiniWoB++, 56 tasks (Table 12) and absolute performance improvement by adding image modality to HTML input for WebGUM (Figure 7).

As for Table 12, we refer to Gur et al. (2022) and Humphreys et al. (2022) for the baseline performance. We use 56 tasks as benchmark, while removing some duplicated tasks (e.g. "-nodelay" tasks) from 62 tasks adopted in Gur et al. (2022), which might cause slight difference between the performance presented in this paper and one reported in prior works. During the evaluation on MiniWoB++, we ignore the time limit due to the computational constraints.

Figure 7 presents full results of the absolute performance improvement, subtracting the success rates: `(Success Rate of WebGUM(HTML+Image)) - (Success Rate of WebGUM(HTML))`. The results suggest WebGUM leverages visual inputs for multi-step reasoning tasks with page transitions (e.g. `choose-date-easy` or `-medium`) or the tasks that require global contexts of the page (e.g. `tic-tac-toe` or `grid-coordinate`). See Appendix G for the visualization.

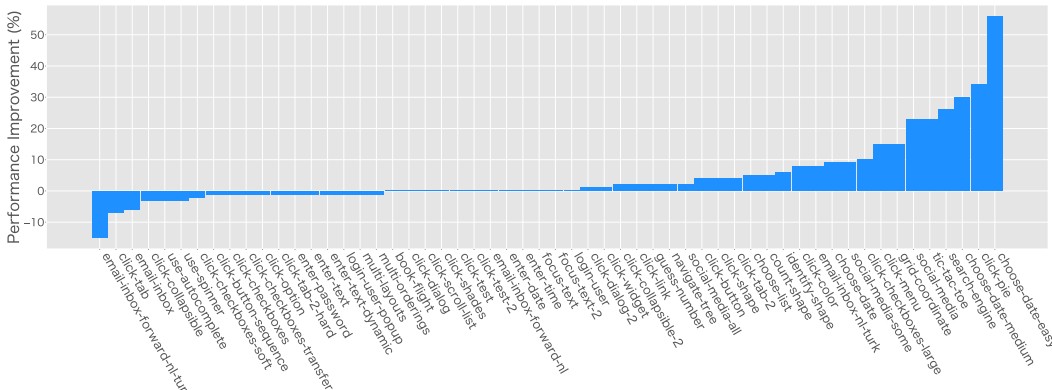

Figure 7: Performance improvement by adding image modality to HTML on 56 tasks from MiniWoB++. We subtract the success rates: `(Success Rate of WebGUM(HTML+Image)) - (Success Rate of WebGUM(HTML))`.

| Task | WebGUM (HTML) | WebGUM (HTML+Image) | WebN-T5 | WGE | CC-Net (SL) | CC-Net (SL&RL) |
|------|---------------|---------------------|---------|-----|-------------|-----------------|
| book-flight | 0.00 | 0.00 | 0.00 | 0.00 | 0.00 | 0.87 |
| choose-date | 0.00 | 0.09 | 0.00 | 0.00 | 0.12 | 0.97 |
| choose-date-easy | 0.08 | 0.64 | 0.03 | – | 0.42 | 0.99 |
| choose-date-medium | 0.04 | 0.34 | 0.00 | – | 0.26 | 0.99 |
| choose-list | 0.16 | 0.21 | 0.26 | 0.16 | 0.19 | 0.99 |
| click-button | 0.96 | 1.00 | 1.00 | 1.00 | 0.78 | 1.00 |
| click-button-sequence | 1.00 | 0.99 | 1.00 | 0.99 | 0.47 | 1.00 |
| click-checkboxes | 1.00 | 0.99 | 0.96 | 0.98 | 0.32 | 0.98 |
| click-checkboxes-large | 0.00 | 0.10 | 0.22 | 0.68 | 0.00 | 0.71 |
| click-checkboxes-soft | 1.00 | 0.98 | 0.54 | 0.51 | 0.04 | 0.95 |
| click-checkboxes-transfer | 1.00 | 0.99 | 0.63 | 0.64 | 0.36 | 0.99 |
| click-collapsible | 1.00 | 0.97 | 0.00 | 1.00 | 0.81 | 1.00 |
| click-collapsible-2 | 0.44 | 0.46 | 0.00 | 0.65 | 0.17 | 0.98 |
| click-color | 0.29 | 0.37 | 0.27 | 1.00 | 0.82 | 1.00 |
| click-dialog | 1.00 | 1.00 | 1.00 | 0.95 | 0.95 | 1.00 |
| click-dialog-2 | 0.32 | 0.33 | 0.24 | 1.00 | 0.88 | 1.00 |
| click-link | 0.98 | 1.00 | 1.00 | 1.00 | 0.59 | 0.99 |
| click-menu | 0.23 | 0.38 | 0.37 | – | 0.22 | 0.94 |
| click-option | 1.00 | 0.99 | 0.37 | 1.00 | 0.21 | 0.99 |
| click-pie | 0.53 | 0.87 | 0.51 | 0.32 | 0.15 | 0.97 |
| click-scroll-list | 0.00 | 0.00 | 0.00 | – | 0.01 | 0.60 |
| click-shades | 0.00 | 0.00 | 0.00 | 0.22 | 0.04 | 1.00 |
| click-shape | 0.60 | 0.64 | 0.53 | 0.64 | 0.11 | 0.95 |
| click-tab | 1.00 | 0.93 | 0.74 | 0.55 | 0.95 | 1.00 |
| click-tab-2 | 0.20 | 0.24 | 0.18 | 0.64 | 0.27 | 0.98 |
| click-tab-2-hard | 0.21 | 0.20 | 0.12 | – | 0.19 | 0.98 |
| click-test | 1.00 | 1.00 | 1.00 | 1.00 | 1.00 | 1.00 |
| click-test-2 | 1.00 | 1.00 | 1.00 | 1.00 | 0.95 | 1.00 |
| click-widget | 0.99 | 1.00 | 1.00 | 0.93 | 0.56 | 1.00 |
| count-shape | 0.64 | 0.69 | 0.41 | 0.59 | 0.21 | 0.85 |
| email-inbox | 0.63 | 0.57 | 0.38 | 0.43 | 0.09 | 1.00 |
| email-inbox-forward-nl | 1.00 | 1.00 | 0.60 | – | 0.00 | 1.00 |
| email-inbox-forward-nl-turk | 0.69 | 0.54 | 0.33 | – | 0.00 | 1.00 |
| email-inbox-nl-turk | 0.46 | 0.54 | 0.23 | 0.77 | 0.05 | 1.00 |
| enter-date | 1.00 | 1.00 | 0.00 | 0.00 | 0.02 | 1.00 |
| enter-password | 1.00 | 0.99 | 0.97 | 0.99 | 0.02 | 1.00 |
| enter-text | 1.00 | 0.99 | 0.89 | 1.00 | 0.35 | 1.00 |
| enter-text-dynamic | 1.00 | 0.99 | 0.98 | 1.00 | 0.39 | 1.00 |
| enter-time | 0.00 | 0.00 | 0.00 | 0.52 | 0.04 | 0.97 |
| focus-text | 1.00 | 1.00 | 1.00 | 1.00 | 0.99 | 1.00 |
| focus-text-2 | 1.00 | 1.00 | 1.00 | 1.00 | 0.96 | 1.00 |
| grid-coordinate | 0.85 | 1.00 | 0.49 | 1.00 | 0.66 | 1.00 |
| guess-number | 0.10 | 0.12 | 0.00 | 0.00 | 0.21 | 1.00 |
| identify-shape | 0.94 | 1.00 | 0.88 | 0.90 | 0.68 | 1.00 |
| login-user | 0.98 | 0.98 | 0.82 | 0.99 | 0.00 | 1.00 |
| login-user-popup | 0.99 | 0.98 | 0.72 | – | 0.02 | 1.00 |
| multi-layouts | 1.00 | 0.99 | 0.83 | 0.99 | 0.00 | 1.00 |
| multi-orderings | 1.00 | 0.99 | 0.88 | 0.99 | 0.00 | 1.00 |
| navigate-tree | 0.98 | 1.00 | 0.91 | 0.99 | 0.32 | 0.99 |
| search-engine | 0.69 | 0.95 | 0.34 | 0.26 | 0.15 | 1.00 |
| social-media | 0.13 | 0.36 | 0.21 | 0.39 | 0.03 | 0.90 |
| social-media-all | 0.00 | 0.02 | 0.00 | 0.01 | 0.00 | 0.75 |
| social-media-some | 0.00 | 0.09 | 0.02 | 0.01 | 0.01 | 0.85 |
| tic-tac-toe | 0.25 | 0.48 | 0.48 | 0.37 | 0.32 | 0.83 |
| use-autocomplete | 0.99 | 0.96 | 0.22 | 0.78 | 0.07 | 1.00 |
| use-spinner | 0.08 | 0.05 | 0.07 | 0.04 | 0.47 | 1.00 |
| **Ave.** | 0.615 | 0.661 | 0.484 | 0.646 | 0.343 | 0.964 |
| **# of Tasks** | 56 | 56 | 56 | 48 | 56 | 56 |

Table 11: Per-task average success rate on 56 tasks from MiniWoB++. Because we omit some duplicated tasks (e.g. "-nodelay" tasks) from 62 tasks adopted in Gur et al. (2022), we recalculate the baseline performances referring Humphreys et al. (2022) and Gur et al. (2022).

# E  COMPOSITIONAL EVALUATION ON MINIWOB++

For the compositional evaluation, we pick up 4 `click-`"something" (link, button, checkboxes, dialog) tasks and make some combinations of those by naively stitching with 2 or 3 tasks. Then, we prepare the following 6 combinational tasks,

- `click-button click-checkboxes`
- `click-button click-dialog`
- `click-button click-link`
- `click-link click-button`
- `click-link click-button click-dialog`
- `click-link click-dialog`

These tasks should be resolved in order of the name: for instance, in `click-link click-button click-dialog` task, the agent should click the proper link, click the proper button, click the proper dialog, and then the task results in successc. In `click-button click-link` task, the agent should click the proper button, and then click the proper link. The instructions for compositional tasks are also simply combined among original task instructions in order of the name. This evaluation could test the ability to transfer primitive skills to control computers to solve unseen tasks.

Table 12 shows the per-task average success rate among 6 combinations above. Interestingly, our multimodal WebGUM achieves significantly better performance (44.0%) on the combination of 3 tasks, i.e. `click-link click-button click-dialog`, compared to WebN-T5 (8.0%) and WebGUM with HTML inputs (4.0%).

| Compositional Task | WebN-T5 (Gur et al., 2022) | WebGUM (HTML) | WebGUM (HTML+Image) |
|---|---|---|---|
| click-button click-checkboxes | 0.26 | 0.46 | 0.89 |
| click-button click-dialog | 0.95 | 0.85 | 0.41 |
| click-button click-link | 0.87 | 0.64 | 0.31 |
| click-link click-button | 0.35 | 0.91 | 0.96 |
| click-link click-button click-dialog | 0.08 | 0.04 | 0.44 |
| click-link click-dialog | 0.55 | 0.80 | 0.80 |
| **Ave.** | 0.510 | 0.617 | 0.635 |

Table 12: Per-task average success rate on 6 tasks from compositional MiniWoB++.

## F  EVALUATION ON WEBSHOP

In addition to MiniWoB++, we extensively evaluate our WebGUM on WebShop (Yao et al., 2022a) benchmark, an online-shopping websites simulator with a large amount of real-world product data. WebShop provides user instruction that describes the feature of items (e.g. *I need a long clip-in hair extension which is natural looking, and price lower than 20.00 dollars*). The agents should search, compare and choose a proper product that matches the given instruction. Since this requires complex multi-step reasoning considering previous contexts for comparison, we can test the capability of instruction-finetuned LLM in decision making tasks in depth. The performance score is evaluated by the percentage of required attributes covered by the chosen product (from 0 to 100), and if the product meets all the requirements, that episode is labeled a success.

Because WebShop does not have API to get the screenshot of rendered websites, we focus on WebGUM with text inputs, parsed from noisy HTML in the real world.[8] We convert the actions from raw texts (e.g. `search[a long clip-in hair extension]` or `click[<item id>]`) to dictionary-like format (e.g. `{"action": "search", "ref": "a long clip-in hair extension"}` or `{"action": "click", "ref": "<item id>"}`), as we use in MiniWoB++, to improve the prediction accuracy. We finetune Flan-T5-XL with about 1K human demonstrations curated by Yao et al. (2022a)[9], using only high-score demonstrations. The score threshold is `score ≥ 50` and we have 840 episodes in total (Table 14). We construct the model input with action history, instruction, and text observation, the same as MiniWoB++ experiments. We evaluate our method with 500 user instructions in the test set.

Table 13 shows that WebGUM achieves 45.0% success, significantly outperforming not only simple baselines, such as supervised imitation learning (IL) and IL plus RL-finetuing (by more than 15%), but also recent prompt-based LLM agents, including ReAct (Yao et al., 2022b) (i.e. PaLM-540B (Chowdhery et al., 2022) with one-shot prompt and reasoning annotations), while our model only has 3 billion parameters. IL and IL plus RL-finetuning baselines use BART (Lewis et al., 2019) model for the search policy, and BERT (Devlin et al., 2019) model for the click policy. The better performance of WebGUM strengthens the observations that instruction-finetuned language models are beneficial even for decision making problems as well as common NLP tasks.

| Methods | Training | Model | Modality | Score | Success Rate |
|---|---|---|---|---|---|
| Rule | – | – | Text | 45.6 | 9.6% |
| IL | SL | BART, BERT | Text+Image | 59.9 | 29.1% |
| IL+RL | SL+RL | BART, BERT | Text+Image | 62.4 | 28.7% |
| Act | In-context | PaLM-540B | Text | 62.3 | 30.1% |
| ReAct | In-context | PaLM-540B | Text | 66.6 | 40.0% |
| WebN-T5 | SL | T5-XL | Text | 61.0 | 29.8% |
| WebGUM | SL | Flan-T5-XL | Text | **67.5** | **45.0%** |
| Human | – | – | Text+Image | 82.1 | 59.6% |

Table 13: Average score and success rate on WebShop (Yao et al., 2022a) benchmark. WebGUM based on Flan-T5-XL achieves 45.0% success, outperforming most baseline approaches including ReAct, a prompted PaLM-540B with reasoning annotations. We refer Yao et al. (2022b) for the baselines.

| Threshold | # of Episodes | Score | Success Rate |
|---|---|---|---|
| `score ≥ 0` | 1026 | 67.2 | 44.4% |
| `score ≥ 50` | 840 | **67.5** | **45.0%** |
| `score = 100` | 497 | 65.3 | 44.4% |

Table 14: Average score and success rate on WebShop with different score thresholds. Because we should balance the dataset size and proficiency, we choose 50 as a threshold.

---

[8]WebShop just provides visual features of item pictures when the agents reach the product page. These features are extracted by ResNet-50 (He et al., 2016), rather than raw images or screenshots of the website. Some baseline agents (IL and IL+RL) incorporate such embeddings.

[9]`https://github.com/princeton-nlp/WebShop/tree/master/baseline_models/data`

# G    EXAMPLE EPISODES OF WEBGUM

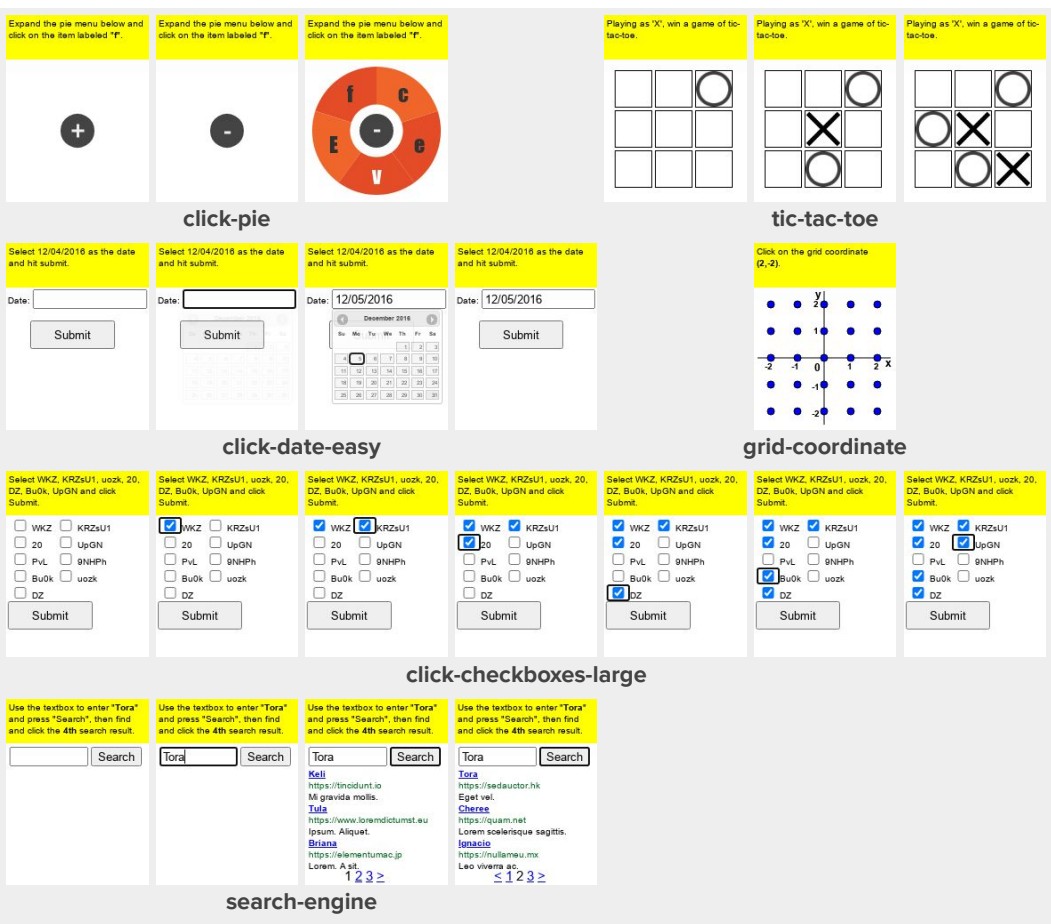

Figure 8: Example of successful episodes demonstrated by multimodal WebGUM on MiniWoB++ (Shi et al., 2017; Liu et al., 2018). The time step goes from left to right. We pick up the tasks which performance is improved by adding image modality. As discussed in Section 5.1, image modality seems to be leveraged for multi-step reasoning tasks with some page transitions (e.g. `click-date-easy`, `click-checkboxes-large`, `search-engine`) or tasks that require global visual contexts (e.g. `click-pie`, `tic-tac-toe`, `grid-coordinate`).

| | |
|---|---|
| **Instruction:** i am looking for dark denim color ethylene vinyl ultra train of size 10, 3rd generation for men, and price lower than 160.00 dollars | **Instruction:** i would like a xlarge plus red camellia fleece jacket that can be machine washed, and price lower than 130.00 dollars |
| **Action:** search [dark denim color ethylene vinyl ultra train of size 10, 3rd generation for men] 
 **Observation:** 
 [back to search] page 1 (total results: 50) [next] 

 [b08b1mcfkl] 
 salewa ultra train 3 - men's 
 $104.0 to $182.99 
 [b07j5mc2x1] 
 skechers men's low-top trainers, us men 
 $53.39 to $140.35 
 [b07x5hr3tz] 
 bruno marc men's linen canvas stretch loafer shoes 
 $39.99 

 **Action:** click [b08b1mcfkl] 
 **Observation:** 
 [back to search] [prev] 
 size [7] [7.5] [8] [8.5] [9] [9.5] [10] [10.5] [11] [11.5] [12] [12.5] [13] 
 color [ombre blue | red orange] [raw green | black out] [dark denim | red orange] 
 salewa ultra train 3 - men's 
 price: $104.0 to $182.99 
 rating: n.a. 
 [description] [features] [reviews] [buy now] 

 **Action:** click [dark denim | red orange] 
 **Observation:** you have clicked dark denim | red orange. 

 **Action:** click [10] 
 **Observation:** you have clicked 10. 

 **Action:** click [buy now] 
 **Termination:** Your score (min 0.0, max 1.0): 1.0 | **Action:** search [xlarge plus red camellia fleece jacket that can be machine washed] 
 **Observation:** 
 [back to search] page 1 (total results: 50) [next] 

 [b09h2qsj5p] 
 men's thermal underwear pants usb heated warm cotton with zone 3 heating long trousers leggings bottoms 
 $56.77 
 [b09hcddf1x] 
 men's thermal underwear pants usb heated warm cotton long trousers leggings bottoms for winter washable baselayer 
 $56.77 to $56.99 
 [b09jgmr43j] 
 premium soft and smooth faux fur loose plush vest jackets fashion sexy fuzzy sleeveless outwear 
 $9.47 to $12.47 

 **Action:** click [next] 
 **Observation:** 
 [back to search] page 2 (total results: 50) [prev] [next] 

 [b07cqt8rbl] 
 king and queen write hoodie couple matching sweater pullover hooded sweatshirt jacket 
 $26.99 to $29.99 
 [b09jmvfpzg] 
 mens sherpa quilted lined flannel jacket plus size plaid button down fleece shirts coats thicken warm cotton outwear 
 $36.79 to $55.99 
 [b09kxfgbn2] 
 mankalvxy womens jacket cardigan flannel cowl neck asymmetric loose outerwear plus size xs-xxl 0-22 
 $20.3 to $26.85 

 **Action:** click [b07cqt8rbl] 
 **Observation:** 
 [back to search] [prev] 
 size [king-small] [queen-small] [king-medium] [queen-medium] [king-large] [queen-large] [king-xxl] [king-xlarge] [queen-xlarge] [queen-xxlarge] 
 king and queen write hoodie couple matching sweater pullover hooded sweatshirt jacket 
 price: $26.99 to $29.99 
 rating: n.a. 
 [description] [features] [reviews] [buy now] 

 **Action:** click [prev] 
 **Observation:** 
 [back to search] page 2 (total results: 50) [prev] [next] 

 [b07cqt8rbl] 
 king and ... 
 [b09jmvfpzg] 
 mens sherpa ... 
 [b09kxfgbn2] 
 mankalvxy womens... 

 **Action:** click [b09jmvfpzg] 
 **Observation:** 
 [back to search] [prev] 
 color [army green] [black] [coffee] [gray] [green] [red] [a-army green] [a-red] 
 size [small] [medium] [large] [x-large] [xx-large] [3x-large] 
 mens sherpa quilted lined flannel jacket plus size plaid button down fleece shirts coats thicken warm cotton outwear 
 price: $36.79 to $55.99 
 rating: n.a. 
 [description] [features] [reviews] [buy now] 

 **Action:** click [red] 
 **Observation:** you have clicked red. 

 **Action:** click [x-large] 
 **Observation:** you have clicked x-large. 

 **Action:** click [buy now] 
 **Termination:** Your score (min 0.0, max 1.0): 1.0 |

Table 15: Example of successful episodes demonstrated by WebGUM on WebShop (Yao et al., 2022a). WebGUM could search and choose the required product and choose proper options (left). In addition, WebGUM could also compare the products with browsing and backtracking (i.e. clicking "next" or "prev" buttons) during the episodes (right).

