# OpenReview forum: "Instruction-Finetuned Foundation Models for Multimodal Web Navigation"
_ICLR.cc/2023/Workshop/RRL — RRL 2023 Poster_

### Official Review · Reviewer_j59d · 2023-02-17
**Excellent paper with extensive experiments and ablation studies.**

**Rating:** 4
**Confidence:** 2

**Review:**

The paper proposes a multimodal agent for web navigation. The agent is based on a vision transformer and a language model. The vision transformer is used to extract image features from web page screenshots. The language model is used to encode HTML documents. The vision and language features are concatenated and used to predict web actions. The authors show that their method outperforms previous approaches on the MiniWoB and WebShop benchmarks. The paper presents extensive experiments and ablation studies. The paper is well written and easy to follow. Tables and figures are clear and easy to understand. The authors are also releasing a large dataset accompanying the paper.

Pros:
- Very well written paper.
- Extensive numerical experiments and ablation study.

Cons:
- Some clarification about the novelty of the paper would be helpful.
- The proposed method is still far away from the SOTA Humphreys et al. (2022) on the WebShop benchmark.

I recommend acceptance.

## Comments:

### 1. Introduction

- To help the reader locate the paper in the context of the field, it would be helpful to clarify the "novel" aspects of WebGUM in a distinct paragraph. Specifically, the unexpert reader will benefit from a brief description of which components of WebGUM (image+HTML multimodality, instruction-finetuning, etc.) are novel and which have already been proposed in previous work.

### 5. Results

#### 5.1. Does Image Modality Help for Task Success?

- The impact of adding the image modality does not seem to be very significant in Table 2. What does "single" mean in the table? The reader will benefit if that information is provided in the caption.

---

### Official Review · Reviewer_esyf · 2023-02-28
**A good read and well structured experimental study using pretrained models for sequential decision making.**

**Rating:** 4
**Confidence:** 4

**Review:**

In our honest opinion, the authors provide a very well written and structured paper, showing great methodical skills in their experimental study.

They first introduce the overall problematic of autonomous web navigation, then present how prior work has tackled related tasks as well as the shortcomings of such approaches when it comes to solving the task of Web Navigation. Subsequently, they present their approach, WebGUM, leveraging (1) Instruction-Finetuned Language Models, augmented with (2) Vision Transformers for multimodal data processing, and (3) trained on a huge amount of multimodal behavioural dataset of web interactions, which they collected using Large Language Models trained on human demonstrations. While the model in this approach is optimised in order to fit/solve the task at hand, this isn't done from scratch as it uses pretrained modules (i.e. reuses prior computation). As a result, we feel that this fits well within the scope of the present venue.

Additionally, we find that the authors expose clearly expose the research questions guiding their methodology, thus proceeding with adequately tackling each of them. By means of an ablation study, they evaluate each component of their approach and experimentally show that :
(i) the addition of visual information, the use of historical observations (2 images) does provide an increase in performance on the task ;
(ii) instruction-finetuned LLMs outperforms unsupervised LLMs and transfer well from standard NLP tasks to the context of multimodal sequential decision making ;
(iii) the performance of their method has a steep increase with the dataset size, while the increase is slightly milder with the model size (i.e. number of parameters) ;
(iv) their approach has higher robustness against some perturbations to the HTML code than those based on prior finetuned and unimodal LLMs.

-----

Overall, the approach seems sound and rigorously executed, with the exception of two points which we believe the authors could either clarify or improve on in order to complete this work :

(1) Lines 253-256 : "Then, we train other models with this dataset and use them for data collection again. We run those models with 10,000 episodes per task and discard failure cases." In this statement, it is unclear to us why the negative examples are being discarded and not used to provide the agent with a negative reward, for instance.

(2) Lines 268-270 : "Due to the huge computational requirements, we run one seed to train each model throughout
the paper. " In this statement, while it is clear that the burden of limited computational resources might have hindered the ability to evaluate the approach on several seeds, we believe this to be a considerable shortcoming as it reduces the statistical significance of the results.

-----

We thank the authors for the great read.